# Potential economic and clinical implications of improving access to snake antivenom in five ASEAN countries: A cost-effectiveness analysis

**Chanthawat Patikorn[1,2], Ahmad Khaldun Ismail[3], Syafiq Asnawi Zainal Abidin[4], Iekhsan Othman[4], Nathorn Chaiyakunapruk[2,5,6]\*, Suthira Taychakhoonavudh◉[1]\***

**1** Department of Social and Administrative Pharmacy, Faculty of Pharmaceutical Sciences, Chulalongkorn University, Bangkok, Thailand, **2** Department of Pharmacotherapy, College of Pharmacy, University of Utah, Salt Lake City, Utah, United States of America, **3** Department of Emergency Medicine, Faculty of Medicine, Universiti Kebangsaan Malaysia, Jalan Yaacob Latif, Bandar Tun Razak, Kuala Lumpur, Malaysia, **4** Jeffrey Cheah School of Medicine and Health Sciences, Monash University Malaysia, Jalan Lagoon Selatan, Bandar Sunway, Selangor, Malaysia, **5** IDEAS Center, Veterans Affairs Salt Lake City Healthcare System, Salt Lake City, Utah, United States of America, **6** School of Pharmacy, Monash University Malaysia, Selangor, Malaysia

\* Nathorn.Chaiyakunapruk@utah.edu (NC); Suthira.T@chula.ac.th (ST)

**Data Availability Statement:** The data that support the findings of this study are available from the

## Abstract

### Background

Despite domestic production of antivenoms in the Association of Southeast Asian Nations (ASEAN) countries, not all victims with snakebite envenomings indicated for antivenom received the appropriate or adequate effective dose of antivenom due to insufficient supply and inadequate access to antivenoms. We aimed to conduct a cost-effectiveness analysis to project the potential economic and clinical impact of improving access to antivenoms when all snakebite envenomings in ASEAN countries were hypothetically treated with geographically appropriate antivenoms.

### Methodology

Using a decision analytic model with input parameters from published literature, local data, and expert opinion, we projected the impact of "full access" (100%) to antivenom, compared to "current access" in five most impacted ASEAN countries, including Indonesia (10%), Philippines (26%), Vietnam (37%), Lao PDR (4%), and Myanmar (64%), from a societal perspective with a lifetime time horizon. Sensitivity analyses were performed.

### Principal findings

In base-case analyses, full access compared to current access to snake antivenom in the five countries resulted in a total of 9,362 deaths averted (-59%), 230,075 disability-adjusted life years (DALYs) averted (-59%), and cost savings of 1.3 billion USD (-53%). Incremental cost-effectiveness ratios (ICERs) of improving access to antivenom found higher outcomes but lower costs in all countries. Probabilistic sensitivity analyses of 1,000 iterations found that 98.1–100% of ICERs were cost-saving.

corresponding authors, ST and NC, upon reasonable request.

**Funding:** This work is supported by the Wellcome Trust [218539/Z/19/Z] to CP, AKI, IO, SAZA, ST, and NC (https://wellcome.org). This research project is supported by the Second Century Fund (C2F), Chulalongkorn University to CP and ST (https://c2f.chula.ac.th). The funders had no role in study design, data collection, data analysis, data interpretation, writing of the report, or the decision to submit for publication.

**Competing interests:** The authors have declared that no competing interests exist.

## Conclusion/Significance

Improving access to snake antivenom will result in cost-saving for ASEAN countries. Our findings emphasized the importance of further strengthening regional cooperation, investment, and funding to improve the situation of snakebite victims in ASEAN countries.

## Author summary

In the Association of Southeast Asian Nations (ASEAN) countries, it was estimated that annually there were 242,648 snakebite victims in ASEAN of which 15,909 victims were dead. Despite domestic production of antivenoms in ASEAN countries, not all victims with snakebite envenomings indicated for antivenom received the appropriate or adequate effective dose of antivenom due to insufficient supply and inadequate access to antivenoms. Especially in Indonesia, Philippines, Vietnam, Lao PDR, and Myanmar, where 4–64% of victims who needed antivenoms were treated with antivenoms. We conducted a cost-effectiveness analysis to project the potential economic and clinical impact of improving access to antivenoms when all victims with snakebite envenomings in the five most impacted ASEAN countries were hypothetically treated with geographically appropriate antivenoms. Improving access to snake antivenom to the full level of access compared to the current level in the five ASEAN countries resulted in a total of 9,362 deaths averted (-59%), 230,075 disability-adjusted life years (DALYs) averted (-59%), and cost savings of 1.3 billion USD (-53%). Our study demonstrated improving access to snake antivenom from the current to the full level of access in ASEAN countries is a cost-saving strategy. Our findings emphasized the importance of further strengthening regional cooperation, investment, and funding to improve the situation of snakebite victims in ASEAN countries to reach the ultimate goal where all victims with snakebite envenoming needing antivenom adequately received the geographically appropriate antivenoms.

## Introduction

Snakebite is a highly prioritized neglected tropical disease recognized by the World Health Organization (WHO). Due to the high global burden of snakebite, WHO has set its goal to reduce morbidity and mortality of snakebite by 50% by 2030 [1–3]. WHO has developed four strategic objectives to tackle the problems, including empowering and engaging communities, ensuring safe and effective treatment, strengthening health systems, and increasing partnerships, coordination, and resources [3].

The Association of Southeast Asian Nations (ASEAN) is an economic union comprising of ten member countries including Brunei Darussalam, Cambodia, Indonesia, Lao PDR, Malaysia, Myanmar, Philippines, Singapore, Thailand, and Vietnam with over 600 million population [4]. ASEAN is among the tropical regions with a disproportionately high incidence of snakebite. Our previous study estimated that there were approximately 243,000 snakebite victims with 16,000 deaths and 950 amputations from snakebite envenoming in seven ASEAN countries, namely, Malaysia, Thailand, Indonesia, Philippines, Vietnam, Lao PDR, and Myanmar. The annual economic and disease burden of snakebite in these countries was estimated at approximately 2.5 billion US Dollars (USD) and 392,000 disability-adjusted life years (DALYs) lost due to snakebite [5].

Previous economic evaluations have demonstrated the cost-effectiveness of antivenoms over no treatment for victims who suffered from snakebite envenoming indicating that antivenoms should be included as part of the pharmaceutical benefits schemes [6–8]. Antivenoms are already included in the essential medicine lists in many countries in the ASEAN [9]. However, not all victims with snakebite envenoming in ASEAN countries could access to geographically appropriate antivenoms for many reasons including inadequate supplies of antivenom, inefficient supply chain system, and inappropriate treatment seeking behavior [9]. Especially in Indonesia, Philippines, Vietnam, Lao PDR, and Myanmar, where 4–64% of victims with snakebite envenoming were treated with antivenoms [5].

Lack of access to antivenom in ASEAN countries could actually be avoided with evidence-informed strategies to improve access to snake antivenom with the goal that every victim with snakebite envenoming should receive antivenoms. However, strategies used to achieve this are likely to be complex and different across countries depending on each country's context and situation. Moreover, improving access to antivenoms could not be solely done by increasing the production of antivenoms. It requires a multifaceted approach involving strengthening the whole system surrounding the management of snakebite victims, such as accurate informatics, rigorous regulations of antivenoms, efficient supply chain, rational use of antivenoms, appropriate treatment seeking behaviors, and good governance to support a strong healthcare system. To accelerate the development of strategies to improve access to snake antivenoms, it is needed to demonstrate the potential impact of treating all victims with snakebite envenoming with snake antivenoms.

Therefore, we aimed to conduct a cost-effectiveness analysis to project the potential economic and clinical impact of improving access to snake antivenoms when all victims with snakebite envenomings in ASEAN countries were hypothetically treated with geographically appropriate antivenoms. Our findings would emphasize the unmet medical needs of snakebite victims in ASEAN countries and the importance of developing strategies to provide access to geographically appropriate antivenoms for all victims with snakebite envenoming to reduce the burden of snakebite in the region.

## Methods

An economic evaluation was conducted using a decision analytic model to assess the cost-effectiveness of improving access to snake antivenom from the current level to full access in ASEAN countries. We projected the economic and clinical implications of "full access" to antivenom relative to "current access" in a hypothetical cohort of snakebite victims in each country from a societal perspective with a lifetime time horizon to capture lifetime costs and consequences of snakebite victims. We developed our study following the methodological considerations for economic evaluations of snakebites described in the previous systematic review [6]. We reported our study following the Consolidated Health Economic Evaluation Reporting Standards (CHEERS) 2022 statement [10].

### Setting

We selected the five most impacted ASEAN countries for this study, including Indonesia, Philippines, Vietnam, Lao PDR, and Myanmar because the previous estimates in these countries found that only 4–64% of victims who were indicated for antivenoms were treated with antivenoms [5].

Malaysia and Thailand were not selected because more than 90% of victims in these countries who were indicated for antivenoms were treated with antivenoms [5]. Brunei Darussalam and Singapore were not selected because snakebite rarely occurs and/or exact data were

lacking [1]. Cambodia was not selected due to a lack of information and key informants, although it is one of the countries with a high incidence of snakebites [1].

## Decision analytic model

A decision analytic model (**Fig 1**) was adapted from the previously developed model to estimate the number of snakebite victims occurring in one year in ASEAN countries and the economic and disease burden of snakebite victims [5]. Briefly, victims who were bitten by snakes sought either conventional treatment or traditional treatment. The victims who were indicated for antivenom treatment might be treated with antivenom depending on the level of access. Snakebite victims might be alive, alive with disabilities, or dead. Disabilities included in this model were digit and limb amputation.

There were four key assumptions of the model [5]. First, one person can be bitten by a snake only once in a lifetime. Second, snakebite victims were accompanied by relatives or family members who took care of them during the snakebite episode. Third, due to lack of data, antivenom effectiveness was based on a study in Nigeria which found 2.33 folds (95% confidence interval; 1.26–4.06) increased risk of death in antivenom indicated victims who were not treated with antivenom compared to those treated with antivenom [11]. Fourth, current access to antivenom was determined as the proportion of the number of antivenoms treatment available by a total number of victims indicated for antivenom treatment with the values of 0.04 (Lao PDR), 0.10 (Indonesia), 0.26 (Philippines), 0.37 (Vietnam), and 0.64 (Myanmar) that were previously estimated [5]. Full access was modeled as all snakebite victims who were indicated for antivenom could be treated with geographically appropriate antivenoms. In the full access scenario, all snakebite victims who firstly sought traditional healers when access is now full are assumed to switch to conventional treatment.

## Input parameters

Input parameters for each country (**S1 Table**) were based on the previous study that estimated the economic and disease burden of snakebite in ASEAN countries [5]. These parameters were sought from published literature, data from the Ministry of Health in each country, local data, and expert opinion [11–32]. The input parameters were validated by triangulation of data from the literature, local data, and interview with key informants who were experts in snakebite in ASEAN.

Main sources of information were national statistics and published research for the burden estimation of Malaysia, Thailand, and Myanmar. Published research and anecdotal evidence (local data, and expert opinion) were the main sources of information for the burden estimation of Vietnam and Lao PDR. Anecdotal evidence was the main source of information for the burden estimation of Indonesia, and Philippines.

## Costs

Costs of snakebite in this model included direct costs and indirect costs (**S1 Table**) [5]. Direct costs included costs of hospitalization, antivenom treatment, antivenom logistics, adverse drug reaction management, amputation, transportation, and additional food for victims and their relatives. Costs of antivenom treatment were estimated based on the average dose of antivenom vials used in the treatment of snakebite envenoming with the consideration of different types of snakes and antivenoms in each country. Indirect costs included productivity losses during snakebite episodes of victims and their relatives and productivity losses due to premature death. Productivity losses due to premature death were discounted at the rate of 3% and adjusted for the annual growth of GDP per capita in each country [29–32].

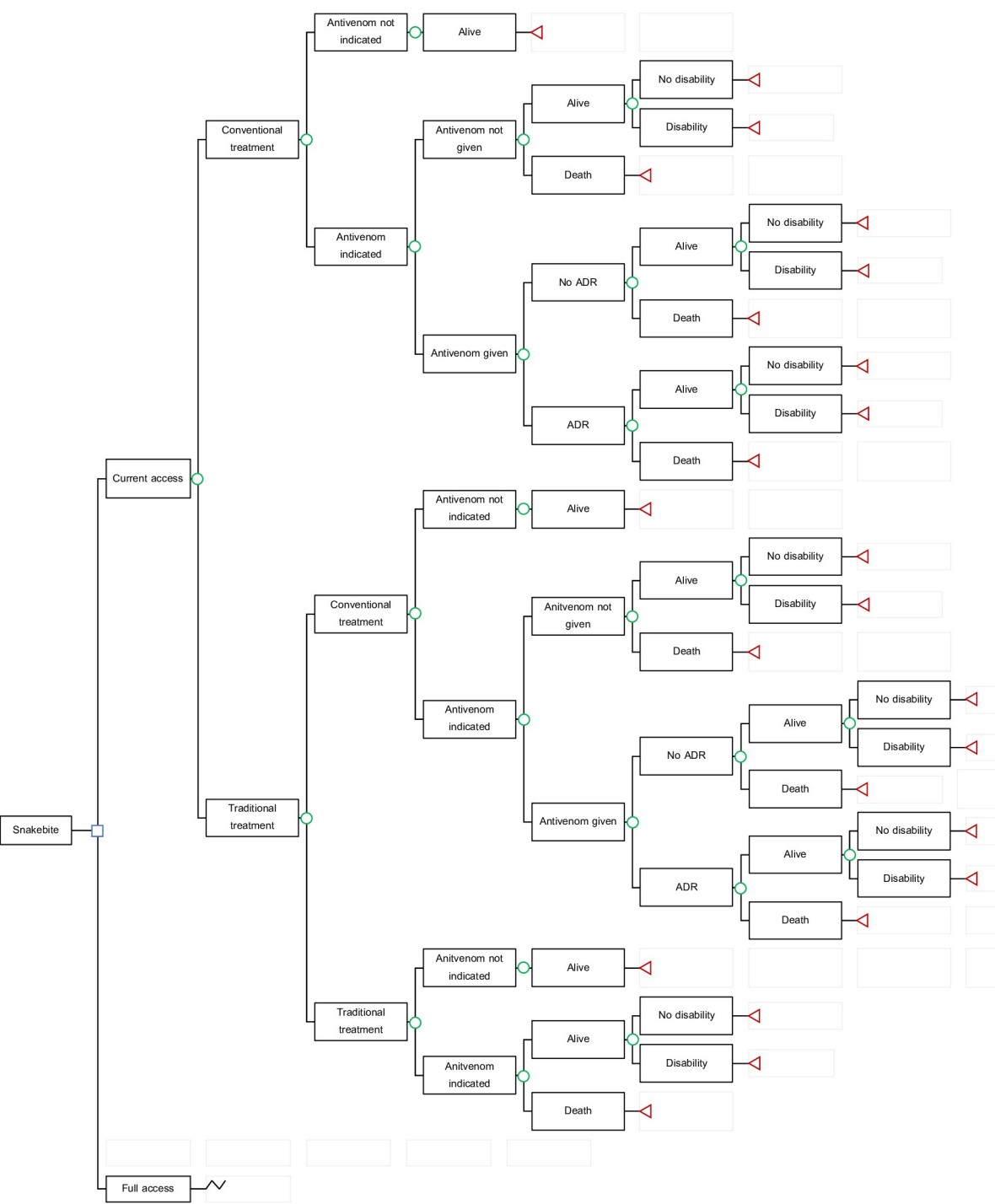

**Fig 1. Decision analytic model to estimate economic and disease burden of snakebite in ASEAN countries.** Abbreviation: ADR–adverse drug reaction.

## Health outcomes

Health outcomes of the model included the number of deaths from snakebite envenoming and disability-adjusted life years (DALYs) lost due to snakebite. DALYs were estimated using the

template developed by WHO [33]. DALYs were the sum of years of life lost (YLL) and years lived with disability (YLD). YLLs due to snakebite envenoming were calculated from the number of deaths multiplied by a global standard life expectancy at the age of death. YLDs of snakebite victims included YLDs for snakebite episodes and YLDs for amputations. YLDs were calculated from the duration of disability multiplied by a disability weight for each condition according to the Global Burden of Disease 2013 study (**S1 Table**) [19].

## Discounting

Costs and health outcomes that occurred after one year were discounted at the rate of 3% [29,30].

## Base-case analyses

In the base-case analyses, the expected costs and outcomes for each level of access were calculated. Primary outcomes of the model were deaths averted, DALYs averted, and incremental costs of full access compared to current access. Costs were expressed in 2019 US Dollars (USD) which equaled to 14,147.67 Indonesian Rupees, 51.80 = Philippine Pesos, 23,050.24, Vietnamese Dong, 8,679.41 Lao Kip, and 1,518.26 Myanmar Kyat [34].

The results were presented as incremental cost-effectiveness ratio (ICER) per death averted and ICER per DALY averted in each country. The willingness-to-pay (WTP) thresholds were based on the local pharmacoeconomic guidelines in Indonesia (4,136 USD) [30], set by the Formulary Executive Council in the Philippines (2,317 USD) [35], and based on the country's GDP per capita in countries without explicit WTP thresholds including Vietnam (2,715 USD), Lao PDR (2,625 USD), and Myanmar (1,421 USD) [31]. ICERs per death averted and ICER per DALY averted with values below these WTP thresholds were considered cost-effective.

## Sensitivity analyses

A series of sensitivity analyses were performed to evaluate the robustness of the base-case conclusions. One-way sensitivity analyses were performed to assess the impact of varying input parameters from minimum to maximum values on the ICERs.

We performed a series of scenario analyses. In the base-case analyses, antivenom had no effect on amputation. Thus, we performed scenario analyses by assuming that antivenom treatment could reduce the amputation rate of snakebite victims with the same relative risk of 2.33 in antivenom indicated victims who were not treated with antivenom compared to those treated with antivenom [11]. Scenario analyses were performed by incorporating post-traumatic stress disorder (PTSD) in the model as a mental disability occurring in 8% of victims who survived snakebite envenoming [5,36–38]. We performed scenario analyses by excluding indirect costs from the model. Lastly, scenario analyses were performed by increasing the logistic costs from 5% to 10% of antivenom price because these could be higher, especially in archipelagic countries like Indonesia and Philippines.

Threshold sensitivity analyses were performed to estimate the lowest level of antivenom effectiveness and the highest level of costs of antivenom treatment that would result in the "not cost-saving" situation where ICER is equal to zero but still considered cost-effective.

Probabilistic sensitivity analyses were performed to assess the model robustness and uncertainty of the base case input parameters over their plausible ranges on the model output. Monte Carlo simulations for 1,000 iterations of ICERs were performed by randomly sampling all input parameters based on the probability distributions. Results of probabilistic sensitivity analyses were presented using the cost-effectiveness plane and the cost-effectiveness acceptability curves (CEACs).

## Results

### Base-case analyses

We projected the economic and clinical impact of improving access to snake antivenom in ASEAN countries from the societal perspective (**Table 1**). When compared to current access, full access to antivenom in each country could save 433 lives in Philippines to 5,981 lives in Indonesia and reduce 10,473 DALYs in Philippines to 148,684 DALYs in Indonesia. However, full access to antivenom resulted in a higher number of patients with amputations (4 amputees in Philippines to 122 amputees in Indonesia) since more patients are being treated and as a result, more survivors from snakebite envenoming. Full access to antivenom had higher direct costs (2 million USD in Philippines to 50 million USD in Indonesia) but less indirect costs (-1,091 million USD in Indonesia to -28 million USD in Philippines) when compared to current access which resulted in total cost savings of 27 million USD in Philippines to 1,040 million USD in Indonesia. In total, when compared to current access, full access to snake antivenom in ASEAN countries resulted in 9,362 deaths averted (-59%), 230,075 DALYs averted (-59%), and cost savings of 1.3 billion USD (-53%).

Base-case analyses of ICERs per death averted and ICERs per DALY averted of full access compared to current access found higher outcomes (0.02 deaths averted in Vietnam to 0.06 death averted in Lao PDR and 0.5 DALYs averted in Vietnam to 1.2 DALYs averted in Myanmar) but lower costs (-7,698 USD in Indonesia to -1,370 USD in Myanmar) in all five ASEAN countries. Thus, improving access to snake antivenom will result in cost-saving (**Table 2**).

### Sensitivity analyses

One-way sensitivity analyses were presented with tornado diagrams to show the percentage change of base-case ICERs corresponding to varying values of the input parameters (**S1** and **S2 Figs**). The most influential parameters for ICERs per death averted were discount rate and relative risk of death when antivenoms are not available. The most sensitive parameters for ICERs per DALY averted were relative risk of death when antivenoms are not available, discount rate, and probability of death in snakebite victims treated with antivenom. One-way sensitivity analyses found that the base-case conclusions were robust.

**Table 1. Estimated economic and clinical impact of snakebite victims between current access and full access to snake antivenom in ASEAN countries.**

|  | Indonesia | | | Philippines | | | Vietnam | | | Lao PDR | | | Myanmar | | |
|---|---|---|---|---|---|---|---|---|---|---|---|---|---|---|---|
|  | Current | Full | Difference | Current | Full | Difference | Current | Full | Difference | Current | Full | Difference | Current | Full | Difference |
| **Health outcomes** | | | | | | | | | | | | | | | |
| Deaths, n | 10,547 | 4,566 | -5,981 | 550 | 117 | -433 | 1,655 | 619 | -1,037 | 1,007 | 141 | -866 | 2,145 | 1,099 | -1,046 |
| Amputations, n | 799 | 921 | +122 | 12 | 16 | +4 | 0 | 0 | 0 | 141 | 202 | +60 | 0 | 0 | 0 |
| DALYs for snakebite | 262,888 | 114,203 | -148,684 | 13,317 | 2,844 | -10,473 | 40,250 | 15,111 | -25,139 | 24,532 | 3,513 | -21,019 | 50,830 | 26,070 | -24,759 |
| **Economic burden, million USD** | | | | | | | | | | | | | | | |
| Direct costs | 59 | 109 | +50 | 1 | 2 | +2 | 7 | 16 | +9 | 0.1 | 1 | +1 | 4 | 11 | +7 |
| Indirect costs | 1,931 | 840 | -1,091 | 83 | 54 | -28 | 261 | 100 | -161 | 80 | 12 | -69 | 75 | 39 | -36 |
| Total costs | 1,990 | 950 | -1,040 | 83 | 57 | -27 | 268 | 116 | -152 | 81 | 13 | -68 | 79 | 50 | -29 |

Costs are presented as USD where 1 USD = 14,147.67 Indonesian Rupees = 51.80 Philippine Pesos = 23,050.24 Vietnamese Dong = 8,679.41 Lao Kip = 1,518.26 Myanmar Kyat. DALY—disability-adjusted life year; USD–US Dollar.

**Table 2. Cost-effectiveness of improving access to snake antivenom in ASEAN countries.**

| | Costs, USD | Deaths, n | DALYs | Incremental costs, USD | Deaths averted, n | DALYs averted | Incremental costs per Death averted, USD | Probability of being cost-saving[*] | Incremental costs per DALY averted, USD | Probability of being cost-saving[*] | WTP Threshold, USD per DALY averted |
|---|---|---|---|---|---|---|---|---|---|---|---|
| **Indonesia** | | | | | | | | | | | |
| Current access (reference) | 14,733 | 0.08 | 1.9 | | | | | | | | |
| Full access | 7,035 | 0.03 | 0.8 | -7,698 | 0.04 | 1.1 | Cost-saving | 99.8% | Cost-saving | 100% | 4,136 |
| **Philippines** | | | | | | | | | | | |
| Current access (reference) | 6,223 | 0.04 | 1.0 | | | | | | | | |
| Full access | 1,536 | 0.01 | 0.2 | -4,687 | 0.03 | 0.8 | Cost-saving | 99.9% | Cost-saving | 99.9% | 2,317 |
| **Vietnam** | | | | | | | | | | | |
| Current access (reference) | 5,733 | 0.04 | 0.9 | | | | | | | | |
| Full access | 2,473 | 0.01 | 0.3 | -3,260 | 0.02 | 0.5 | Cost-saving | 99.9% | Cost-saving | 100% | 2,715 |
| **Lao PDR** | | | | | | | | | | | |
| Current access (reference) | 5,620 | 0.07 | 1.7 | | | | | | | | |
| Full access | 910 | 0.01 | 0.2 | -4,710 | 0.06 | 1.5 | Cost-saving | 100% | Cost-saving | 100% | 2,625 |
| **Myanmar** | | | | | | | | | | | |
| Current access (reference) | 3,760 | 0.10 | 2.4 | | | | | | | | |
| Full access | 2,390 | 0.05 | 1.2 | -1,370 | 0.05 | 1.2 | Cost-saving | 98.1% | Cost-saving | 98.1% | 1,421 |

[*]Percentage of 1,000 iterations that were cost-saving based on a probabilistic sensitivity analysis. Costs are presented as 2019 USD where 1 USD = 14,147.67 = Indonesian Rupees = 51.80 = Philippine Pesos = 23,050.24 Vietnamese Dong = 8,679.41 Lao Kip = 1,518.26 Myanmar Kyat. DALY–Disability-adjusted life year; USD–US Dollars; WTP–Willingness-to-pay.

Scenario analyses were presented in **S2** and **S3 Tables**. Incorporating PTSD in the model, assuming that antivenom could reduce the risk of amputation, and increasing the logistic costs of antivenom resulted in similar ICERs per death averted and ICERs per DALY averted in all countries indicating that improving access will result in cost-saving. Excluding indirect costs from the model resulted in ICERs per death averted ranging from 1,488 in Lao PDR to 8,632 USD per death averted in Vietnam. These ICERs per death averted were above the WTP thresholds in all countries except Lao PDR. While ICERs per DALY averted excluding indirect costs from the model ranged from 61 in Lao PDR to 356 in Vietnam. These ICERs per DALY averted were below the WTP thresholds in all countries.

Threshold analyses of antivenom effectiveness and costs of antivenom treatment resulted in an ICER of 0 are shown in **S4 Table**. The lowest level of antivenom effectiveness, presented as a risk ratio of death in indicated victims who were not treated with antivenom compared to those treated with antivenom, was ranging from 0.35 in Lao PDR to 1.19 in Myanmar. The highest level of costs of antivenom treatment was ranging from 9 to 149 times the base-case value in Myanmar and Lao PDR, respectively.

Probabilistic sensitivity analyses were performed for 1,000 iterations with the results presented in **Figs 2** and **S2** and **S3**. We found that 98.1–100% of 1,000 ICERs per death averted and 98.3–100% of 1,000 ICERs per DALYs averted were cost-saving.

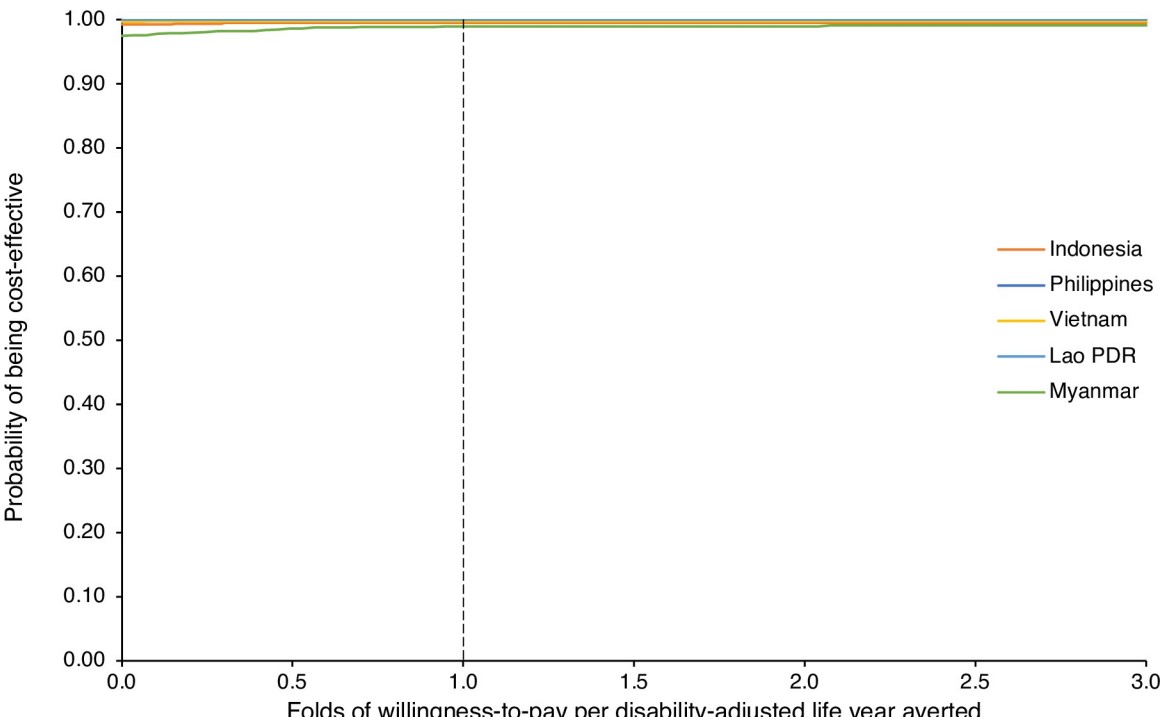

**Fig 2. Cost-effectiveness acceptability curves of improving access to snake antivenom in ASEAN countries.** Probabilities of full access to snake antivenom being cost-effective compared to current access in each ASEAN country at varying willingness-to-pay thresholds based on 1,000 iterations of a probabilistic sensitivity analysis are presented.

## Discussion

This is the first study to our understanding that projected the cost-effectiveness of improving access to snake antivenom. This cost-effective analysis was not done to evaluate whether antivenom was cost-effective or not because antivenoms were already available in ASEAN but not all victims with snakebite envenoming were treated antivenoms [5,9]. Thus, we tried to evaluate the potential economic and clinical impacts of increasing the access to antivenom in ASEAN countries when all snakebite victims in the full access scenario are treated with geographically appropriate antivenom. We used a decision analytic model with input parameters from various sources including published literature, local sources, and expert opinion. We did not propose specific strategies to improve access to snake antivenoms as each country has different problems which require different strategies and policies to address them [9]. Rather, we demonstrated the potential economic and clinical impact when all victims with snakebite envenoming in ASEAN countries were treated with geographically appropriate antivenoms.

We found that improving access to snake antivenoms would result in cost-saving with higher outcomes (deaths and DALYs averted) but lower costs. In total, when compared to current access, full access to snake antivenom in five ASEAN countries resulted in 9,362 deaths averted (-59%), 230,075 DALYs averted (-59%), and cost savings of 1.3 billion USD (-53%). Although full access to antivenom compared to current access had higher direct costs because all victims with snakebite envenoming received antivenom treatment, the direct costs were entirely offset by indirect costs because antivenoms could save the victims' lives which would avoid tremendously productivity losses due to premature death.

Mortality and disabilities of snakebite envenoming in each country differ in regards to differences in the toxicity and lethality of the snakes [5,9]. Nevertheless, improving access to

antivenom was found to be cost-saving in all five ASEAN countries regardless of differences in snakes causing the snakebite envenoming and baseline level of access to antivenom. We performed a series of sensitivity analyses and found that the conclusion of our study remained robust. Threshold analyses found that the antivenom effectiveness could be as low as 0.35 to 1.19, and the costs of antivenom treatment could increase as high as 9 to 149 times of the base-case value to render improving access to antivenom no longer a cost-saving strategy. This emphasizes the cost-effectiveness of improving access to antivenom.

ASEAN has made significant progress in the management of snakebite and antivenom, but there remain challenges in this region to be addressed especially the lack of snakebite-related informatics system and inadequate access to antivenoms [9]. Our previous estimates highlighted the high burden of snakebite in ASEAN despite the availability of domestically produced antivenoms. It was estimated that there were approximately 243,000 snakebite victims with 16,000 deaths and 950 amputations from snakebite envenoming in these countries with the estimated annual economic and disease burden of snakebite of approximately 2.5 billion USD and 392,000 DALYs lost due to snakebite [5]. Findings of this study indicated that improving access to antivenoms in ASEAN countries would result in tremendous cost savings for the whole society. Thus, further investment and funding is warranted so that we could achieve the WHO's goal to halve the snakebite burden in ASEAN [3].

This study supports and informs that improving access to snake antivenom where all victims with snakebite envenoming received geographically appropriate antivenoms will result in cost-saving. As a result, policy makers and relevant stakeholders in ASEAN to develop effective strategies to improve access to antivenom and reduce the burden of snakebite victims in the region given that antivenom is a lifesaving drug that should be universally accessible. However, improving access to antivenom is not only about increasing the production of antivenoms or purchasing more antivenoms, but also strengthening the whole health system to effectively deal with snakebite problems. More importantly, encouraging and engaging communities is needed to change the behavior of snakebite victims to seek care at appropriate healthcare facilities instead of traditional healers or seek no care at all. We previously discussed the potential opportunities to improve access to antivenom in ASEAN that included accurate estimation of antivenom demand, rigorous regulations of antivenom, strengthening the supply chain system, raising public awareness about the importance of treating snakebite envenoming by healthcare professionals, strengthening the health system to ensure appropriate snakebite management and rational use of antivenoms, and expanding collaboration of local and international stakeholders to better improve access to snake antivenom for victims in the region [9]. Nevertheless, there is no single strategy that could improve access to snake antivenom in every country. Strategies should be developed with consideration of the actual challenges and barriers to policy implementation in individual countries such as the infrastructure and capacity of the health system to appropriately tackle the snakebite problem.

## Limitations

There were limitations in our approach that needed to be discussed. Firstly, although Cambodia is one of the countries with snakebite victims, Cambodia was not included in this study because of a lack of published literature and key informants. However, given our findings in five ASEAN countries, improving access to antivenom would be highly cost-saving as well. Secondly, all input parameters used in the model were derived from the available published literature, local data, and expert opinion when data were not available. These input parameters carried inherent uncertainty. We assumed antivenom effectiveness based on a study in Nigeria [11]. Most snakebites In Nigeria are inflicted by snakes of the genus *Echis*, for which

antivenoms, in general, have high effectiveness [7]. This assumption may have a limitation as snakes in Nigeria are different from ASEAN. Nevertheless, our sensitivity analyses showed robust conclusions of the model. We strongly emphasized the need to conduct comprehensive research to estimate the true burden of snakebite in ASEAN. Thirdly, other disabilities of snakebite envenoming such as blindness, malignant ulcers, and pregnancy loss were not included due to lack of empirical evidence [7]. Chronic kidney disease due to Russell's viper (*Daboia russelii*) bite was not included in our study. It was found that Russell's viper bite caused acute kidney injury. However, information on chronic kidney disease following Russell's viper bite in ASEAN was not documented because patients were lost to follow-up after they were discharged [39–41]. This emphasizes the importance of funding future studies in ASEAN to evaluate all relevant consequences and disabilities and associated costs of snakebite to allow better estimation of the cost-effectiveness of improving access to antivenom. Fourthly, we assumed that all snakebite victims in full access would eventually seek conventional treatment. However, this might not be possible because not all victims could have timely access to healthcare facilities, especially those who lived in the farthest rural areas. This is especially important as timely access to healthcare facilities is related to the prognosis of the victims, both in terms of mortality and long-term disability [42,43]. Lastly, costs of strategy to improve access of antivenoms from the current level to full e.g., costs of increasing antivenom manufacturing capacity, or costs of improving the supply chain, were not included in the analysis. However, these costs were assumed to be covered by the costs of antivenom treatment.

## Conclusion

Our study demonstrated improving access to snake antivenom from the current to the full level of access in five ASEAN countries was cost-saving. Our findings indicated that the WHO's goal to halve the snakebite burden could be achieved by providing full access to snake antivenoms for all victims in ASEAN which emphasized further strengthening regional cooperation, investment, and funding to improve the situation of snakebite victims in ASEAN countries to reach the ultimate goal where all victims with snakebite envenoming needing antivenom adequately received the geographically appropriate antivenoms.

## Supporting information

**S1 Appendix. CHEERS 2022 Checklist.**
(DOCX)

**S1 Table. Input parameters for economic evaluation of improving access to snake antivenom in ASEAN countries.**
(DOCX)

**S2 Table. Sensitivity analysis of cost-effectiveness analysis of improving access to snake antivenom in ASEAN countries in different scenarios.**
(DOCX)

**S3 Table. Sensitivity analysis of cost-utility analysis of improving access to snake antivenom in ASEAN countries in different scenarios.**
(DOCX)

**S4 Table. Threshold analyses of antivenom effectiveness and costs of antivenom treatment resulted in an incremental cost-effectiveness ratio of 0.**
(DOCX)

**S1 Fig. One-way sensitivity analysis of incremental costs per death averted of improving access to snake antivenom in ASEAN countries.**
(DOCX)

**S2 Fig. One-way sensitivity analysis of incremental costs per disability-adjusted life year (DALY) averted of improving access to snake antivenom in ASEAN countries.**
(DOCX)

**S3 Fig. Cost-effectiveness plane of incremental costs per death averted of improving access to snake antivenom in ASEAN countries.** Incremental costs and deaths averted of full access to snake antivenom in each ASEAN country based on a probabilistic sensitivity analysis of 1,000 iterations are presented in dots. Willingness-to-pay thresholds of each country are presented as dash lines with corresponding color.
(DOCX)

**S4 Fig. Cost-effectiveness plane of incremental costs per disability-adjusted life year (DALY) averted of improving access to snake antivenom in ASEAN countries.** Incremental costs and disability-adjusted life years (DALYs) averted of full access to snake antivenom in each ASEAN country based on a probabilistic sensitivity analysis of 1,000 iterations are presented in dots. Willingness-to-pay thresholds of each country are presented as dash lines with corresponding color.
(DOCX)

## Acknowledgments

The authors would like to thank members of the Pan ASEAN Antivenom (PAAV) consortium for their information and insightful recommendations on this work.

## Author Contributions

**Conceptualization:** Chanthawat Patikorn, Nathorn Chaiyakunapruk, Suthira Taychakhoonavudh.

**Data curation:** Chanthawat Patikorn.

**Formal analysis:** Chanthawat Patikorn, Nathorn Chaiyakunapruk, Suthira Taychakhoonavudh.

**Funding acquisition:** Ahmad Khaldun Ismail, Iekhsan Othman, Nathorn Chaiyakunapruk, Suthira Taychakhoonavudh.

**Investigation:** Chanthawat Patikorn, Iekhsan Othman, Nathorn Chaiyakunapruk, Suthira Taychakhoonavudh.

**Methodology:** Chanthawat Patikorn, Nathorn Chaiyakunapruk, Suthira Taychakhoonavudh.

**Project administration:** Chanthawat Patikorn, Syafiq Asnawi Zainal Abidin, Iekhsan Othman, Nathorn Chaiyakunapruk, Suthira Taychakhoonavudh.

**Resources:** Chanthawat Patikorn, Ahmad Khaldun Ismail, Iekhsan Othman, Nathorn Chaiyakunapruk, Suthira Taychakhoonavudh.

**Validation:** Chanthawat Patikorn, Ahmad Khaldun Ismail, Iekhsan Othman, Nathorn Chaiyakunapruk, Suthira Taychakhoonavudh.

**Visualization:** Chanthawat Patikorn, Suthira Taychakhoonavudh.

**Writing – original draft:** Chanthawat Patikorn, Nathorn Chaiyakunapruk, Suthira Taychakhoonavudh.

**Writing – review & editing:** Chanthawat Patikorn, Ahmad Khaldun Ismail, Syafiq Asnawi Zainal Abidin, Iekhsan Othman, Nathorn Chaiyakunapruk, Suthira Taychakhoonavudh.

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
