## [Decision Letter · Decision Letter 0]

21 Sep 2022

Dear Miss Taychakhoonavudh,

Thank you very much for submitting your manuscript "Potential economic and clinical implications of improving access to snake antivenom in ASEAN countries: a cost-effectiveness analysis" for consideration at PLOS Neglected Tropical Diseases. As with all papers reviewed by the journal, your manuscript was reviewed by members of the editorial board and by several independent reviewers. In light of the reviews (below this email), we would like to invite the resubmission of a significantly-revised version that takes into account the reviewers' comments. 

Since there are major methodological issues have been identified in the study, results and conclusions also affect by those issues. Hence it is mandatory to consider each of the stated issues in the methodology and revise the results and conclusions sections. The specific issues which could not be revised the adapted, those issues should be carefully described as limitations under the discussion. The details of how those limitations affects the conclusions should be described.

We cannot make any decision about publication until we have seen the revised manuscript and your response to the reviewers' comments. Your revised manuscript is also likely to be sent to reviewers for further evaluation.

Sincerely,

Kalana Prasad Maduwage, MBBS, MPhil, PhD, FRSPH (UK)

Academic Editor

Indika Gawarammana

Section Editor

Since there are major methodological issues have been identified in the study, results and conclusions also affect by those issues. Hence it is mandatory to consider each of the stated issues in the methodology and revise the results and conclusions sections. The specific issues which could not be revised the adapted, those issues should be carefully described as limitations under the discussion. The details of how those limitations affects the conclusions should be described.

Reviewer's Responses to Questions

**Key Review Criteria Required for Acceptance?**

**Methods**

-Are the objectives of the study clearly articulated with a clear testable hypothesis stated?

-Is the study design appropriate to address the stated objectives?

-Is the population clearly described and appropriate for the hypothesis being tested?

-Is the sample size sufficient to ensure adequate power to address the hypothesis being tested?

-Were correct statistical analysis used to support conclusions?

-Are there concerns about ethical or regulatory requirements being met?

Reviewer #1: The estimated logistic cost of only 5% is rather low.

Reviewer #2: The objectives of the study are clear and well founded. In this work the authors selected five most impacted ASEAN countries for this study, including Indonesia, Philippines, Vietnam, Lao PDR, and Myanmar. However the title refers to the whole group of ASEAN countries. The title should be modified as to mention that it includes only a group of ASEAN countries, the ones having the highest load of SBE. The methodology is clearly described, and the statistical analyses are presented. There are no ethical issues concerning this study.

I have several comments concerning specific issues in the Methodology:

The disabilities included in the model were digit and lib amputations. Although these disabilities are probably the most significant ones, studies in this topic have identified other types of physical and psychological disabilities which may have a great impact in the quality of life of affected people. In particular, victims of Daboia sp often develop chronic kidney failure, which involves high costs of treatment. This would be a relevant parameter to consider, and there is information in the literature on the incidence of this problem and the incidence of Daboia sp cases, at least in some of these countries or in neighbouring countries. Thus, the disability analysis could be expanded. In the section of limitations of the study, the authors indicated that “other disabilities of snakebite envenoming such as blindness, malignant ulcers, and pregnancy loss were not included due to lack of empirical evidence”. However, in the case of chronic renal failure, which involves high costs from the medical system, I believe there is published information which can provide valuable inputs to enrich the analysis.

The authors state that “antivenom effectiveness was based on a study in Nigeria which found 2.33 folds (95% confidence interval; 1.26-4.06) increased risk of death in indicated victims who were not treated with antivenom compared to those treated with antivenom”. Extrapolation of this assumption from Nigeria to South East Asia may have limitations. In Nigeria most snakebites are inflicted by snakes of the genus Echis, for which antivenoms in general have higher efficacy. In the case of Asia, many snakebites are inflicted by species of Naja and Bungarus, which may induce severe neurotoxicity of rapid onset, or by Daboia sp, which inflict highly complicated cases involving severe renal injury and other consequences. Hence, this extrapolation has limitations. It would be more pertinent to find such numbers in studies carried out in India, for example, where the types of snakes are similar to the ones in ASEAN countries.

It is stated that “In the full access scenario, all snakebite victims who firstly sought traditional healers when access is now full are assumed to switch to conventional treatment”. This assumption is valid for the purpose of the study. However, an additional factor is the time needed to reach health facilities, which may largely differ depending on the country and the region, the development of the public health system and circumstances such as the quality of the roads and climate variables, as has been studied for the case of SBE in Nepal. I realize these factors are difficult to introduce in the model used, but at least should be acknowledged in the Discussionm because the time to reach medical attention is directly related to the prognosis of the cases, both in terms of mortality and long term disability..

In estimating the cost of antivenom treatment, the authors have probably based their analyses in the current cost of antivenom in these countries (which I believe vary a lot) and on the usual number of antivenom vials used in a single treatment. However, if the study is based on the idea that the antivenoms to be used are geographically appropriate and effective, both the cost per vial and the number of vials needed for a treatment may vary as compared to the current situation. How are these parameters considered in the analysis?

**Results**

-Does the analysis presented match the analysis plan?

-Are the results clearly and completely presented?

-Are the figures (Tables, Images) of sufficient quality for clarity?

Reviewer #1: It is interesting to know the Budget Impact that each country government (or another payer) needs to pay per year for this antivenom access.

Reviewer #2: Results are presented in a clear manner and the analysis matches the analysis plan described in the methodology. The figures and tables are good and of enough quality and clarity.

**Conclusions**

-Are the conclusions supported by the data presented?

-Are the limitations of analysis clearly described?

-Do the authors discuss how these data can be helpful to advance our understanding of the topic under study?

-Is public health relevance addressed?

Reviewer #1: Yes

Reviewer #2: The main cionclusions are supported by the data presented. However, as indicated above in the comments to the methodology, there are extrapolations in this study which may affect the conclusions. The authors acknowledge some limitations of the study, but should expand this by including other aspects mentioned above.

Since only five countries of ASEAN countries were included in the analysis, it is recommended that the authors acknowledge this limitation and include some considerations on whether this could apply to the whole region or not.

It would be relevant to discuss possible differences between the countries analyzed regarding the snake species that inflict the majority of accidents in each country, the type of clinical picture that they induce and the possible types of sequelae. For example, the incidence of amputation may be higher in countries where bites are predominantly inflicted by snakes having necrotizing effects, such as viperid species and cytotoxic cobras. In contrast, in a country where most cases are inflicted by kraits (Bungarus sp), the likelihood of necrosis and amputation is low, but the likelihood of severe neurotoxic envenoming leading to death is high. On the other hand, in a country where many bites are inflicted by Daboia sp, the possibilities of chronic kidney failure is high. The analysis of these issues, or at least their Discussion, would be enriched with the description of these possible national differences. 

As indicated above in the comments on the methodology, using the data on the reduction of the risk of death from a study in Nigeria, and extrapolating it to ASEAN countries has important limitations that need to be acknowledged. It would be relevant to check whether there is information on mortality due to SBE in India between people treated with antivenom and those who were not treated. At least some information on this regard may indicate whether the extrapolation from data of Nigeria is a valid approximation.

When acknowledging the limitation of including only amputations as disabilities, the authors did not mention the problem of chronic renal failure that often occurs in victims of Daboia sp. This needs to be discussed as well.

**Editorial and Data Presentation Modifications?**

Reviewer #1: (No Response)

Reviewer #2: The manuscript is well written and presented, I do not have editorial suggestions.

**Summary and General Comments**

Reviewer #1: The authors performed a cost-effectiveness analysis of snake antivenom in ASEAN countries. By saving lives and preventing disabilities, the full antivenom access is cost-saving. The result is important for the health care policy changes in these countries. 

1. It is interesting to know the Budget Impact that each country government (or another payer) needs to pay per year for this antivenom access.

2. It is interesting to know the reasons why the antivenom access is limited in each country. Are there major problems in antivenom production and/or logistics? The solution may not be simple by just increasing payment for the antivenom costs. 

3. The problem in each country may be different. Can they produce antivenom by themselves? The antivenom distribution in countries comprising numerous islands, i.e., Philippines and Indonesia, may be difficult. The estimated logistic cost of only 5% is rather low.

Reviewer #2: This study aims to assess the cost-effectiveness of improving access to snake antivenom from current level to full access in ASEAN countries, built on a previous study in which the authors evaluated the burden of SBE in terms of number of cases, mortality and cost. As such, it is a relevant contribution to the field of SBE since this type of analysis is scarce and badly needed. These studies are difficult in part because of the lack of accurate information to feed the models used. Regardless of these limitations, the study clearly demonstrates the high impact that adequate access and use of effective and safe antivenoms would have in ASEAN countries. The authors should consider several issues mentioned in my comments to the methodology since they may contribute to further expanding and improving their analysis. I think tha inclusion of more analyses, along the lines indicated in my comments to the methods, would enrich this work.

PLOS authors have the option to publish the peer review history of their article (what does this mean?). If published, this will include your full peer review and any attached files.

Reviewer #1: No

Reviewer #2: No
---

## [Decision Letter · Decision Letter 1]

28 Oct 2022

Dear Suthira

We are pleased to inform you that your manuscript 'Potential economic and clinical implications of improving access to snake antivenom in five ASEAN countries: a cost-effectiveness analysis' has been provisionally accepted for publication in PLOS Neglected Tropical Diseases.

Best regards,

Kalana Prasad Maduwage, MBBS, MPhil, PhD, FRSPH (UK)

Academic Editor

Indika Gawarammana

Section Editor

Both reviewers are agreed to accept the revised version of the manuscript and I am also agreed with the corrections done. Therefore, as the academic editor, I would like to accept the revised version of the manuscript to be published in PLoS NTD.

Reviewer's Responses to Questions

**Key Review Criteria Required for Acceptance?**

**Methods**

-Are the objectives of the study clearly articulated with a clear testable hypothesis stated?

-Is the study design appropriate to address the stated objectives?

-Is the population clearly described and appropriate for the hypothesis being tested?

-Is the sample size sufficient to ensure adequate power to address the hypothesis being tested?

-Were correct statistical analysis used to support conclusions?

-Are there concerns about ethical or regulatory requirements being met?

Reviewer #1: (No Response)

Reviewer #2: See summary and general comments

**Results**

-Does the analysis presented match the analysis plan?

-Are the results clearly and completely presented?

-Are the figures (Tables, Images) of sufficient quality for clarity?

Reviewer #1: (No Response)

Reviewer #2: See summary and general comments

**Conclusions**

-Are the conclusions supported by the data presented?

-Are the limitations of analysis clearly described?

-Do the authors discuss how these data can be helpful to advance our understanding of the topic under study?

-Is public health relevance addressed?

Reviewer #1: (No Response)

Reviewer #2: See summary and general comments

**Editorial and Data Presentation Modifications?**

Reviewer #1: (No Response)

Reviewer #2: See summary and general comments

**Summary and General Comments**

Reviewer #1: I am satisfied with the revision.

Reviewer #2: Thge authors have carefully and satisfactorily addressed my comments and criticisms to the first version of this manuscript. Some of the limitations I mentioned are due to lack of available information (for example the issue of extrapolating data on mortality from Nigeria and not including the problem of chronic renal failure in the sequelae). The authors have expanded the text on limitations of the study, explaining in more detail why some of my suggestions cannot be answered with the available information.

PLOS authors have the option to publish the peer review history of their article (what does this mean?). If published, this will include your full peer review and any attached files.

Reviewer #1: No

Reviewer #2: No

---

## [Editor Report · Acceptance letter]

2 Nov 2022

Dear Miss Taychakhoonavudh,

We are delighted to inform you that your manuscript, "Potential economic and clinical implications of improving access to snake antivenom in five ASEAN countries: a cost-effectiveness analysis," has been formally accepted for publication in PLOS Neglected Tropical Diseases.

Best regards,

Shaden Kamhawi

co-Editor-in-Chief

Paul Brindley

co-Editor-in-Chief
